# Transport of Alzheimer's associated amyloid-β catalyzed by P-glycoprotein

**James W. McCormick**[1,2‡]\*, **Lauren A. McCormick**[1,3,4‡], **Gang Chen**[1,3], **Pia D. Vogel**[1,3], **John G. Wise**[1,3,4]\*

**1** Department of Biological Sciences, Southern Methodist University, Dallas, Texas, United States of America, **2** Green Center for Systems Biology, University of Texas Southwestern Medical Center, Dallas, Texas, United States of America, **3** The Center for Drug Discovery, Design and Delivery, Southern Methodist University, Dallas, Texas, United States of America, **4** The Center for Scientific Computation, Southern Methodist University, Dallas, Texas, United States of America

‡ Co-first authors on this work.
* jwise@smu.edu (JGW); james.mccormick@utsouthwestern.edu (JWM)

**Data Availability Statement:** All relevant data are within the manuscript and its Supporting Information files.

**Funding:** This work was in part sponsored by an NIH NIGMS grant [R15GM09477102] to JGW and

## Abstract

P-glycoprotein (P-gp) is a critical membrane transporter in the blood brain barrier (BBB) and is implicated in Alzheimer's disease (AD). However, previous studies on the ability of P-gp to directly transport the Alzheimer's associated amyloid-β (Aβ) protein have produced contradictory results. Here we use molecular dynamics (MD) simulations, transport substrate accumulation studies in cell culture, and biochemical activity assays to show that P-gp actively transports Aβ. We observed transport of Aβ40 and Aβ42 monomers by P-gp in explicit MD simulations of a putative catalytic cycle. In *in vitro* assays with P-gp overexpressing cells, we observed enhanced accumulation of fluorescently labeled Aβ42 in the presence of Tariquidar, a potent P-gp inhibitor. We also showed that Aβ42 stimulated the ATP hydrolysis activity of isolated P-gp in nanodiscs. Our findings expand the substrate profile of P-gp, and suggest that P-gp may contribute to the onset and progression of AD.

## Introduction

Alzheimer's disease (AD) is a progressive and irreversible neurodegenerative disease that primarily affects geriatric populations. One of the pathological hallmarks of AD is deposition and accumulation of amyloid-β (Aβ) in the brain, which is thought to be caused not only by increased Aβ production, but also by decreased clearance of Aβ from the brain. [1–3]. The most common Aβ isoform in the brain is the 40-residue peptide Aβ(1–40); however, in certain forms of AD, the 42 residue isoform Aβ(1–42) has also been shown to increase significantly in the brain [4]. Aβ(1–42) is also regarded as the more fibrillogenic of the peptides produced from amyloid precursor protein (APP) degradation [4–6].

There is compelling preclinical evidence that the blood-brain barrier (BBB) plays an important role in Aβ clearance [7]. Aβ efflux across the BBB is a multistep process involving several cofactors, in which the LDL Receptor Related Protein 1 (LRP-1) mediates Aβ uptake at the abluminal surface of brain capillary endothelial cells [8–10]. It has been suggested that Aβ is

by a generous private donation from Ms. Suzy Ruff of Dallas, TX. Undergraduate student support was obtained through the Engaged Learning and Hamilton Undergraduate Research Fellowship programs at Southern Methodist University. There was no additional external funding received for this study.

**Competing interests:** The authors have declared that no competing interests exist.

then transferred from LRP-1 to P-glycoprotein (P-gp, *ABCB1*) in endosomes with the help of PICALM and Rab11, and then actively exported by P-gp at the luminal surface from the endothelium into the blood [8–11]. Although experimental results are mixed, there is evidence that P-glycoprotein actively participates in the efflux of both Aβ(1–42) and Aβ(1–40) (Aβ42 and Aβ40, respectively) [8, 12–20].

P-glycoprotein is a member of the ATP Binding Cassette (ABC) transporter family and effluxes a variety of substrates [18]. The expression of P-gp in brain capillaries is inversely correlated with deposition of Aβ in the brain [8, 21]. Furthermore, endothelial BBB expression of P-gp declines as humans age, and this decrease in P-gp expression is accompanied by reduced functioning of the BBB [22–24]. Taken together, these data suggest an active role of P-gp in Aβ clearance from the brain. This hypothesis is strengthened by comparing cognitively normal brains to age-matched brains of AD patients; the brains of AD patients exhibit significant decreases in P-gp expression and significant increases in Aβ deposition [22, 25]. Mouse models of AD support an active role for P-gp in the transport of AD-associated Aβ [10, 14, 16, 18, 22, 25, 26].

However, *in vitro* studies of interactions between P-gp and Aβ have yielded mixed results. In 2001, Lam et al. observed that Aβ is transported by hamster P-gp [27]. Two subsequent studies, one using human colon adenocarcinoma cells, the other using P-gp transfected porcine LLC cells, support these findings [18, 28]. Inhibition of P-gp in the human hCMEC/D3 cell line resulted in increased intracellular accumulation of Aβ40 [29]. A new study by Chai et al. provides compelling evidence that both Aβ40 and Aβ42 interact with and are transported by P-gp *in vitro* and *ex vivo* [20]. However, a study by Bello & Salerno using paired P-gp overexpressing and non-P-gp-overexpressing human carcinoma lines found that Aβ42 had no effect on the efflux of the P-gp substrate pirarubicin [12]. This study also found that Aβ42 had no effect on the ATPase activity of P-gp in membrane vesicles [12]. Lastly, the overexpression of P-gp in polarized canine MDCK cells did not promote the transcytosis of radiolabeled Aβ40 in transwell assays [11].

In this study, we assessed the ability of P-gp to transport Aβ using both computational and *in vitro* techniques. Using explicit all atom MD simulations, we analyzed and modeled the transport mechanism of Aβ40 and Aβ42 by human P-gp. In biochemical assays, we showed that Aβ42 stimulates the ATPase activity of purified P-gp; however, this stimulation is dependent upon the lipid environment used. In cell culture assays, we observed enhanced intracellular retention of fluorescently-labeled Aβ42 in the presence of Tariquidar, a potent P-gp inhibitor [30]. Our results indicate that Aβ peptides are transport substrates of P-gp, suggesting that P-gp may also be involved in the onset and progression of AD. Understanding the role of P-gp in AD may be of crucial importance for the development of future treatments, and may have implications for compounds targeting P-gp in cancer treatment.

## Results

### Targeted molecular dynamics simulations show transport of amyloid-β by P-gp

P-glycoprotein (P-gp) is an efflux transporter that is highly promiscuous with respect to transport substrates; P-gp has transmembraneous Drug Binding Domains and two cytoplasmic Nucleotide Binding Domains, at which ATP hydrolysis occurs (Fig 1) [31]. Upon binding of a transport substrate and ATP, the NBDs associate and the DBDs transition from open-to-the-inside (cytoplasm) to open-to-the-outside (extracellular space), resulting in both transport of the substrate across the cell membrane and ATP hydrolysis [31]. Both Aβ-42 (4514.04 Da) and Aβ-40 (3429.80 Da) are significantly larger than the largest known substrate of P-gp,

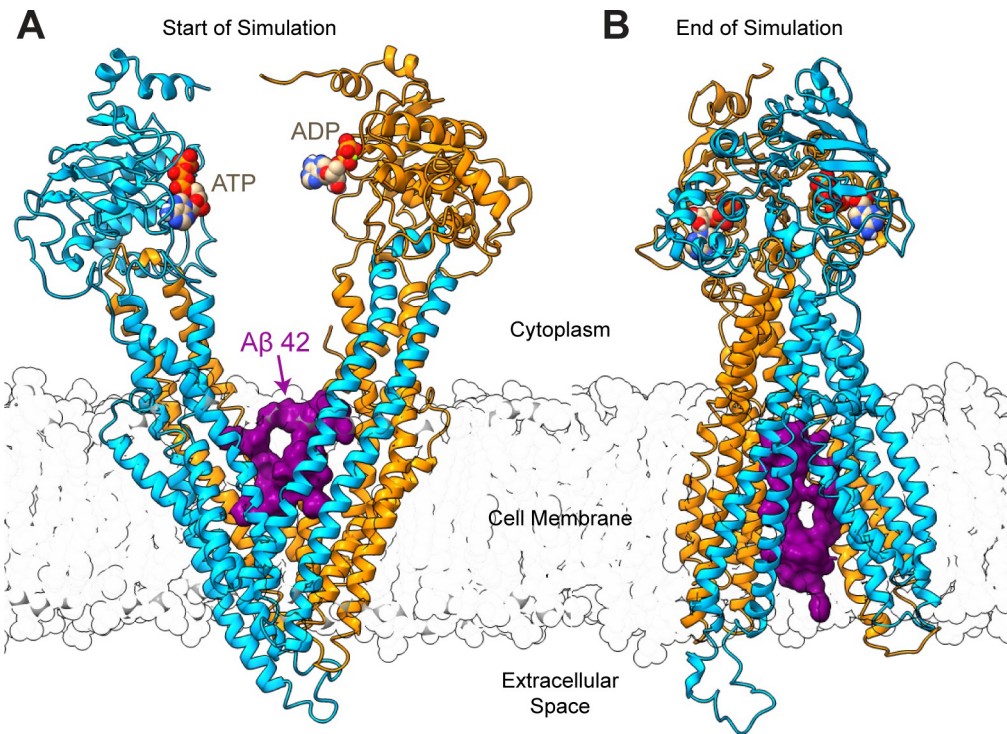

**Fig 1. First and final frames of Targeted Molecular Dynamics (TMD) experiments—P-gp with amyloid-β.** (A) The first frame of a representative simulation of Aβ42 (PDB 1IYT) bound to the drug binding domains of P-gp. (B) The final frame of the same representative TMD simulation shown in (A). The N- and C-terminal halves of P-gp are colored turquoise or orange; Aβ42 is shown in purple surface representation; ATP and ADP are bound at the nucleotide binding domains and shown in van der Waals representation.

cyclosporine A (1,202.61 Da) [32–34]. To investigate whether P-gp is indeed capable of transporting these bulky Aβ peptides, we performed targeted molecular dynamics (TMD) experiments using techniques that have been previously used to study the transport of P-gp substrates [35, 36].

To generate a plausible starting point for TMD simulations, three variants of AD-associated Aβ (Aβ40: PDB IDs 2LFM, 2M4J; Aβ42: PDB ID 1IYT) were docked to human P-gp in a putative starting conformation with the drug binding domains (DBDs) open to the cytoplasm [31, 35, 37–40] (Figs 1 and S1). When docked to the DBDs, each Aβ peptide occupied the previously identified R and H drug binding sites simultaneously, and Aβ-protein contacts were dominated by hydrophobic interactions (S2 Fig and S1 Table) [35, 41]. Our findings suggest that, despite their large sizes, the Aβ peptides are able to fit within the DBDs of P-gp. These data are supported by a separate docking study of Aβ peptides against P-gp performed by Callaghan et al., which used a conformation of P-gp with a slightly less open drug binding region [42]. Despite the decrease in cavity volume, the DBDs of P-gp were able to accommodate Aβ42 or Aβ40.

Furthermore, recent cryo-EM structures of human P-gp bound to the substrate vincristine identified 13 residues within the DBDs that may be important for substrate binding [43]. Specifically, they are M68, M69, F983, Y310, I306, M949, E875, M986, Q946, Q347, F343, and Q990 [43]. In our docking studies of Aβ42 and Aβ40, we observed that the Aβ peptides interacted closely with 11 of the 13 aforementioned residues, the exceptions being F983 and M68. Both F983 and M68 were within 5 Å of the Aβ peptides. Our docking studies support the hypothesis that these residues may be important for substrate binding.

As a potential negative control for our TMD simulations, we mutated every residue in the Aβ42 peptide (1IYT) into arginine, creating polyarginine 42 (P-42). P-42 does not fall into the category of compounds normally transported by P-gp, most of which are hydrophobic [44], but it is similar in molecular weight and size to the tested Aβ peptides (S2 Fig). After assembly of a complete system with Aβ, P-gp, lipids, water and ions, each system was relaxed in unbiased molecular dynamics (MD) simulations [35, 36]. TMD simulations were then performed as described in McCormick et al. 2015 [35, 36]. Briefly, small forces were applied to α-carbons of P-gp to guide the protein through a series of conformational changes, thereby simulating a putative catalytic transport cycle (S1 Fig). Except for the α-carbons of P-gp, no external forces were applied to the Aβ peptides or to any other atoms in the systems.

In each TMD simulation starting with P-gp open to the cytoplasm and with Aβ bound at the DBDs, we observed vectorial movement of Aβ perpendicular to the membrane and towards the extracellular space (n = 6 independent simulations per ligand) (Figs 2 and S2E–S2H). The P-gp–bilayer system was oriented such that the membrane is parallel to the X–Y plane, and movement through the membrane is indicated as movement along the Z-axis. In each set of simulations, we observed movement of the Aβ peptide from the cytoplasmic leaflet of the membrane to the extracellular leaflet of the membrane (Fig 2A–2C).

In these simulations, the center of mass of Aβ40 (derived from 2LFM) was observed to move an average of approximately 7.8 ± 1.3 Å from the cytoplasmic to the extracellular side of P-gp (Fig 2A); Aβ40 (from 2M4J) moved an average of approximately 9.4 ± 1.0 Å (Fig 2B); Aβ42 (from 1IYT) moved an average of approximately 8.4 ± 1.3 Å (Fig 2C). For reference, the previously reported transport of daunorubicin (DAU), a known P-gp substrate, is shown in Fig 2E [35]. Daunorubicin was reported to move an average of 10.0 ± 2.7 Å through the DBDs towards the extracellular space [35]. Movement of the Aβ peptides by P-gp ranged from approximately 8 Å to 10 Å across all simulations.

Although both structures of Aβ40 (2LFM and 2M4J) were moved from the inner to outer leaflet in our simulations, we observed a significant difference between the distance traveled by the two structures of Aβ40 (P = 0.04). There was no significant difference between the distance traveled by Aβ42 and either form of Aβ40 (Aβ40 2LFM vs. Aβ42, P = 0.43; Aβ40 2M4J vs. Aβ42, P = 0.17). Despite the large discrepancy in size (DAU 527.5 g/mol, vs. 4514.04 Da Aβ42), there was no significant difference in the distance traveled by daunorubicin (DAU) and the distance traveled by the Aβ monomers (Aβ42 vs. DAU, P = 0.20; Aβ40 2LFM vs. DAU, P = 0.09; Aβ40 2M4J vs. DAU, P = 0.56). These data suggest that the substrate profile of P-gp may include much larger molecules than was previously thought.

## P-gp does not transport Polyarginine 42 in TMD simulations

Simulations with Polyarginine 42 (P-42) were started at the initial docking pose of Aβ42 (1IYT) (Fig 1D). Fig 2D shows the average distance traveled by the center of mass of P-42 during the transport cycle. In stark contrast to the behavior of the Aβ peptides, P-42 was not moved through the membrane bilayer (n = 6 independent simulations) but remained at a relatively stable position within the DBDs throughout each simulation (0.2 ± 2.6 Å). We observed a highly significant difference between the distance traveled by P-42 and the distance traveled by any of the Aβ monomers, with P < 0.0001 for all three comparisons, respectively (Fig 2F).

## ATPase activity of P-gp in the presence of Aβ42

To test whether Aβ42 interacts directly with P-gp, we used a series of *in vitro* ATPase assays of purified murine P-gp, which has 87% sequence identity and high functional similarity to human P-gp [45]. P-gp exhibits a relatively low rate of ATP hydrolysis in the absence of a

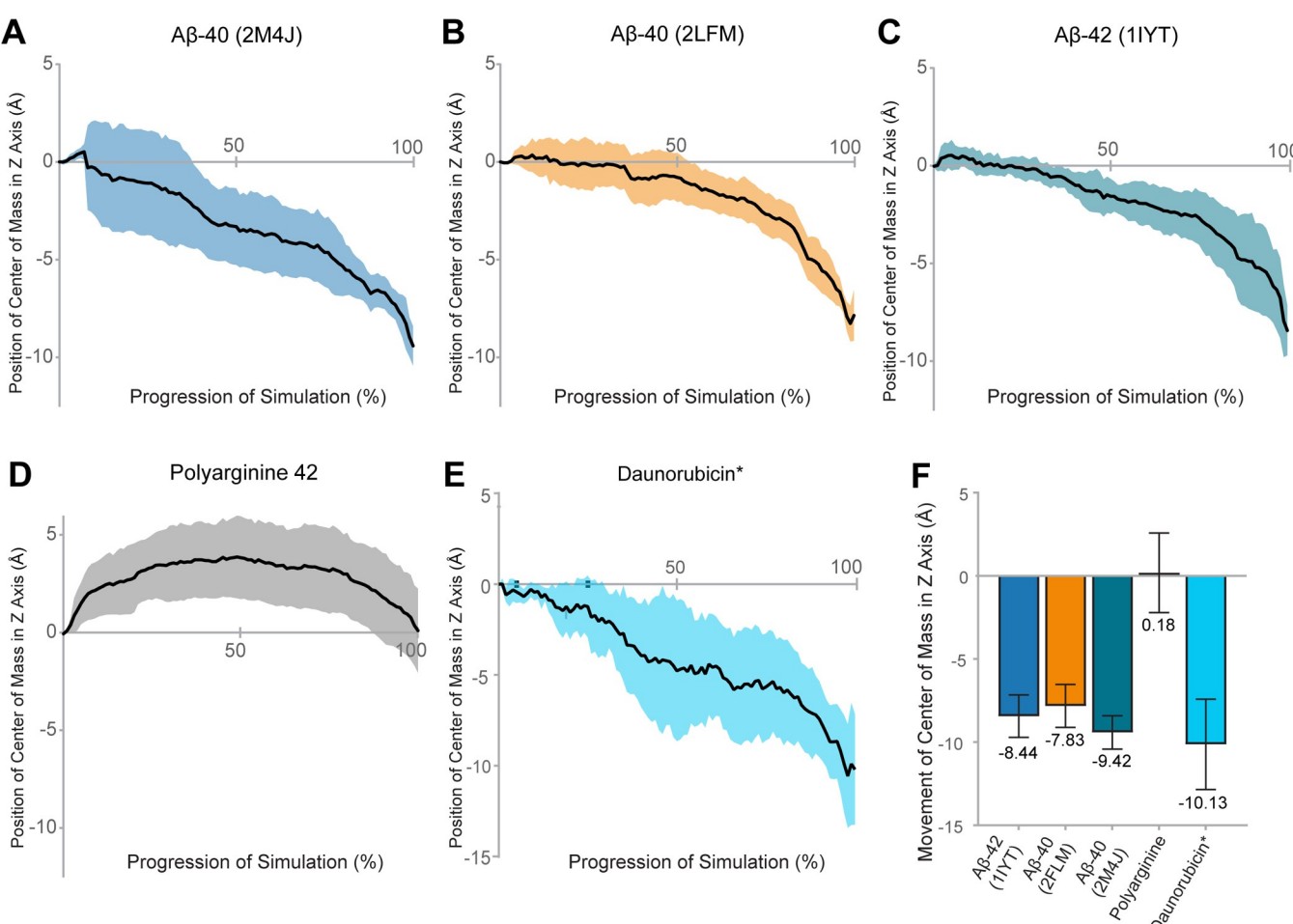

**Fig 2. Movement of amyloid-β by P-gp in molecular dynamics simulations.** The center of mass of (A) Aβ40 structure 2M4J, (B) Aβ40 structure 2LFM, (C) Aβ42 structure 1IYT, (D) Polyarginine 42 peptide and (E) daunorubicin were calculated for each step of the simulated putative catalytic cycle of P-gp. Positional changes were calculated relative to the distance from starting coordinates of the ligand. Data represent the mean position of the center of mass (black line) ± one standard deviation from the mean shown in colored shading, n = 6 simulations per ligand. In these simulations, movement towards the cytoplasm is positive on the Z axis, and movement towards the extracellular space is negative. (F) shows the total mean distance traveled through the plane of the membrane (Z axis) ± one standard deviation. *Data for daunorubicin is reproduced with permission from *McCormick et al. 2015* [35]. Copyright 2015 American Chemical Society.

transport substrate; the introduction of a transport substrate often results in a several fold increase in ATPase activity [46]. Using ATP hydrolysis assays as described in Brewer et al. 2014, we assessed whether monomeric Aβ42 affects the ATPase activity of P-gp [46, 47].

Purified murine P-gp was functionally reconstituted into mixed micelles or lipid bilayer nanodiscs; the latter are considered a more native-like lipid environment [48]. In these studies, 20μg of P-gp in micelles or 15μg of P-gp in nanodiscs were incubated with Aβ42 (molar ratio of 1:18) with or without 150 μM of verapamil (VPL), a substrate of P-gp. A concentration of 150 μM verapamil was selected as this was the amount previously used to obtain maximal activity for murine P-gp expressed in *Pichia pastoris* as reported in Lerner-Marmarosh et al. [49]. In micelles, we report that the basal activity of P-gp was 51 ± 3 nmol/min/mg, and the VPL-stimulated activity was 106 ± 7 nmol/min/mg. In nanodiscs, we report that the basal activity of P-gp was 131 ± 9 nmol/min/mg, and the VPL-stimulated activity was 390 ± 14 nmol/min/mg.

In mixed micelles (Fig 3, blue bars), the ATPase activity of P-gp was stimulated by verapamil (VPL) as expected. Similar to Bello & Salerno 2015, we found that Aβ42 alone did not stimulate the ATPase activity of P-gp in mixed micelles [12]. Furthermore, a combination of VPL and Aβ42 resulted in an increase of ATPase activity above that observed for VPL alone (P < 0.01). However, with P-gp in nanodiscs (Fig 3, orange bars), we found that Aβ42 significantly stimulated the ATPase activity of P-gp (P < 0.005). We observed a 1.42 fold-basal increase in the ATPase activity of P-gp in nanodiscs upon the addition of Aβ42, and these findings closely agree with those of Chai et al., who observed a 1.5 fold-basal increase in activity with P-gp in lipid vesicles upon the addition of 10 µM Aβ42 [20]. Interestingly, we observed that the combination of VPL and Aβ42 did not significantly stimulate ATPase activity relative to VPL alone.

Our data suggest that the effect of Aβ42 on the ATPase activity of P-gp is dependent upon the membrane environment [50, 51]. This could explain the contradictory findings of other ATPase activity studies of Aβ and P-gp; these studies used membrane vesicles derived from a variety of cell types [12, 18, 27]. The stimulation of ATPase activity by Aβ42 indicates that it may interact directly with P-gp, and supports the hypothesis that Aβ42 is a putative transport substrate of P-gp.

## Accumulation of labeled Aβ42 in DU145-TXR and DU145 cells

To test whether we could observe the results of transport of Aβ42 by P-gp in human cellular systems, a fluorescently labeled Aβ42 peptide (HiLyte488-Aβ42) was assayed for accumulation in the paired DU145 and DU145-TXR prostate cancer cell lines [30, 52, 53]. P-gp is significantly over-expressed in the multidrug resistant DU145-TXR cells relative to the parental, chemotherapy sensitive DU145 cells [53]. In contrast to previous studies of Aβ and P-gp using

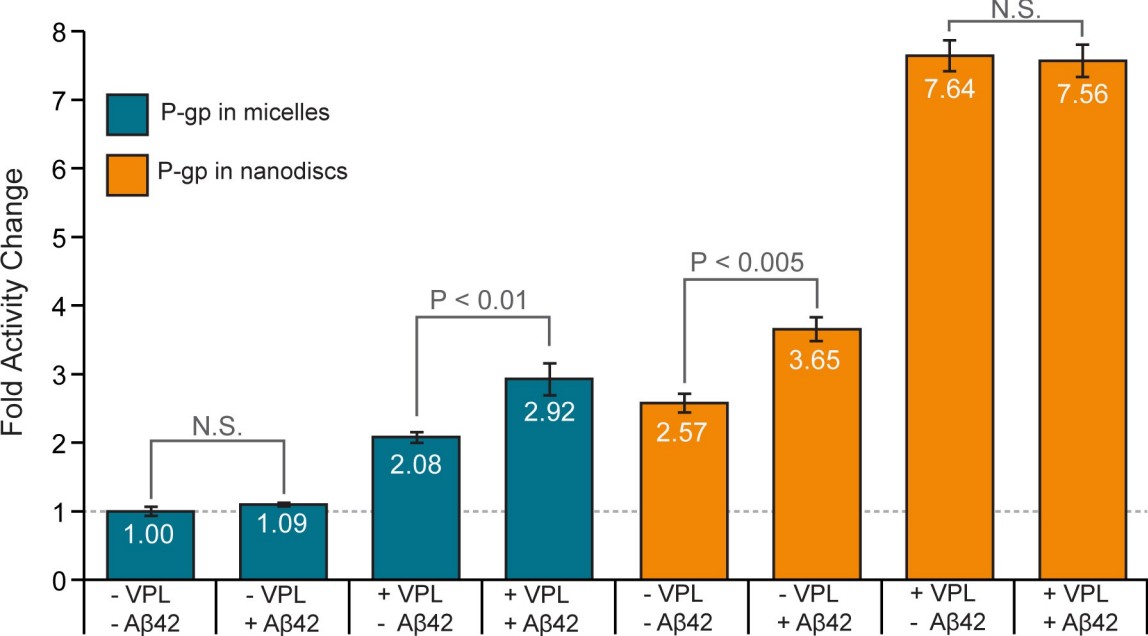

**Fig 3. The effect of amyloid-β 42 on the ATPase activity of P-gp in micelles and nanodiscs.** The effect of verapamil (VPL) and Aβ42 on the rate of ATP hydrolysis by P-gp was measured in micelles and in nanodiscs. All samples are normalized to the basal ATPase rate of P-gp in micelles (blue bars). Error bars represent ± one standard deviation from the mean. For P-gp in micelles, 12.8 µM Aβ42 and 712 nM P-gp were used; for P-gp in nanodiscs, 9.6 µM Aβ42 and 534 nM P-gp were used–both corresponding to a molar ratio of 1:18. A VPL concentration of 150 µM was used, as in [49].

paired cell lines, we used the strong, selective, and non-competitive P-gp inhibitor Tariquidar (TQR) to assess the accumulation of Aβ [12, 18, 27, 30]. Each cell line was treated with 1μM HiLyte488-Aβ42, 1μM of TQR, or a combination of 1μM HiLyte488-Aβ42 and 1μM TQR for 16 hours. The accumulation of HiLyte488-Aβ42 was subsequently quantified using confocal microscopy; Fig 4C–4N shows representative images of each treatment (n = 12 images per trial, two independent trials).

In both the non-P-gp overexpressing and the P-gp overexpressing cell lines, we observed significant increases (P < 0.0001; P < 0.0001) in the accumulation of HiLyte488-Aβ42 in the presence of TQR (Figs 4A and 4B and S4). Although both cell lines showed a TQR-dependent increase in HiLyte488-Aβ42 accumulation, the P-gp overexpressing DU145-TXR cells exhibited the greatest increase in fluorescence (P < 0.0001) (S4 Fig). Increased accumulation of

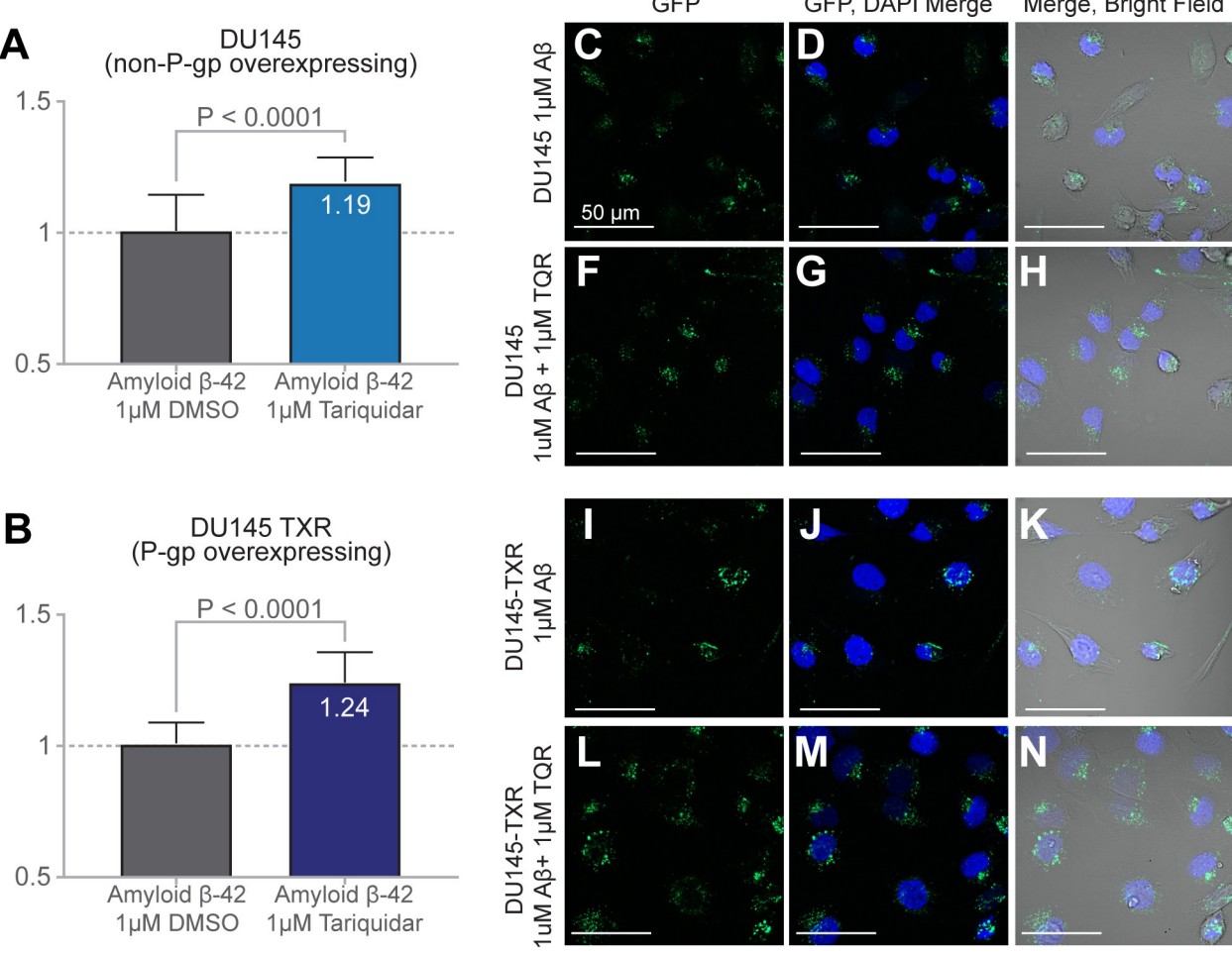

**Fig 4. Intracellular accumulation of HiLyte-488-Aβ42 in DU145 and DU145TXR cells after P-gp inhibition by tariquidar.** The intracellular fluorescence of paired chemotherapeutic sensitive/resistant cancer cell lines (DU145 and DU145-TXR) was measured by confocal microscopy after a 16 hour incubation with fluorescently labeled Aβ42. **(A)** When compared to treatment with 1 μM DMSO control (no added TQR, grey bar), wild type P-gp expressing DU145 cells showed a significant 19% (P < 0.0001) increase in mean intracellular fluorescence when inhibited by 1 μM of the P-gp inhibitor Tariquidar (TQR; blue bar, top panel). **(B)** Compared to treatment with 1 μM DMSO vehicle (no added TQR, grey bar), treatment of P-gp overexpressing DU145-TXR cells showed a significant 24% (P < 0.0001) increase in mean intracellular fluorescence in the presence of 1 μM Tariquidar (purple bar, bottom panel). Representative Images **(C-E)** show DU145 treated with 1 μM Aβ42 alone; **(F-H)** show DU145 treated with 1 μM Aβ42 and 1 μM TQR; **(I-K)** show DU145-TXR cells treated with 1 μM Aβ42 alone; **(L-N)** show DU145-TXR treated with 1 μM Aβ42 and 1 μM TQR.

HiLyte488-Aβ42 upon targeted inhibition of P-gp by TQR indicates that P-gp actively participates in the transport of HiLyte488-Aβ42 in human cellular systems.

## Discussion

The mechanism by which P-gp might transport substrates of such significant size and flexibility as the Aβ peptides remains unclear. Specifically, two primary questions remain: firstly, can the Aβ peptides fit within the DBDs of P-gp, and secondly, are the alternating access motions of P-gp sufficient to move the Aβ peptides across the cell membrane [8]? In answer to the first open question, docking studies performed by us and by Callaghan et al. show that the DBDs of P-gp can easily accommodate the Aβ peptides in both ordered and disordered states [42]. Consequently, the initial association of Aβ with the DBDs of P-gp should be possible. To address the second question, we explored how P-gp might transport Aβ40 and Aβ42 using targeted MD simulations using previously established techniques. Our TMD simulations showed that P-gp is indeed capable of transporting the Aβ peptides through the membrane despite their significant sizes.

In each simulation of the Aβ peptides, we observed vectorial movement of Aβ through the P-gp DBDs and towards the extracellular space, with total movement ranging between 7.8 and 9.4 Å (Figs 2 and S3). These distances correlate well with previously published movements of the P-gp substrate daunorubicin (DAU) in TMD simulations [35]. Interestingly, for both Aβ42 and the 2M4J structure of Aβ40, the bulk of observed movement occurred during the transition between the 3B5X and 2HYD conformations of P-gp, or when the DBDs switch from open-to-the-inside to open-to-the-outside (S1 Fig and S1 Table). In contrast, for both DAU and the 2LFM structure of Aβ40, the bulk of observed movement occurred during the transition from 2HYD to 3B5Z, both of which are open-to-the-outside conformations (S1 Table).

As postulated by Callaghan et al., we observed that the Aβ peptides were indeed 'pinched' during simulated transport by P-gp [42]. However, in our TMD simulations, we found that the transported Aβ peptides were in a more disordered state than that postulated by Callaghan et al. This is unsurprising, since Aβ is partially exposed to the solvent when bound to the inward-open conformation of P-gp, and the helical structures of Aβ have been shown to collapse in water [54]. Since the Aβ peptides are thought to associate with P-gp though a complex network of interactions [8, 20, 55], it is possible that the peptides would contact the solvent prior to entering the DBDs of P-gp, and therefore arrive at the pump in a more disordered state. The partial–or complete–collapse of the ordered helices could further enhance the fit and subsequent transport of these bulky Aβ peptides by P-gp.

Studies have shown that Aβ monomers can fold into structures with two β strands; these β strands allow the monomers to oligomerize and then to potentially assemble into AD-associated amyloid fibrils [56]. At the start of our simulations, the Aβ monomers were not in this folded conformation. It was, however, interesting to ask whether P-gp could somehow facilitate the folding of Aβ monomers during the transport process, and thus contribute to the formation of extracellular amyloid fibrils. In the folded state, both Aβ40 and Aβ42 stabilize the turn between β strands through a salt bridge between Asp23 and Lys28 [57]. In the simulations reported here, this salt bridge was not observed to form for any significant amount of time (S5 Fig). It is possible that hydrophobic interactions with the DBDs prevented the formation of any stable secondary structure by the Aβ peptides (S1 Table). Indeed, residue contacts between the Aβ peptides and P-gp were dominated by hydrophobic, non-polar interactions throughout the transport process, with a notable increase in polar contacts as the DBDs opened to the extracellular space (S8 Fig and S1 Table). Contacts between Aβ and charged residues of the DBDs contributed only a minority of the protein-ligand interactions. Our data suggest that

transport by P-gp may not stabilize or contribute to the folding of Aβ monomers. Since Aβ monomers with a distinct folded structure were not simulated, the ability of P-gp to transport or disrupt folded Aβ monomers is unclear and warrants further study. To date, each study of the effect of Aβ42 upon the ATPase activity of P-gp has used membrane vesicles derived from different cellular systems [12, 20, 27]. Combined with our findings, these data support the growing consensus that the lipid environment strongly affects the behavior and transport activity of P-gp [51, 58–62]. In our ATPase activity assays, we found that monomeric Aβ42 does not stimulate the ATPase activity of P-gp in micelles, but does stimulate the ATPase activity of P-gp in nanodiscs (Fig 3). Therefore, we hypothesize that the differences between our tested systems, and potentially the conflicting results of previous studies, may be due to interactions between P-gp the different lipid environments. Indeed, a new study by Chai et al. shows that both Aβ42 and Aβ40 stimulate the ATPase activity of P-gp in lipid vesicles, providing further evidence that the Alzheimer's associated Aβ peptides interact directly with P-gp, and may be transport substrates of P-gp [20].

Our ATPase studies suggest that Aβ42 interacts directly with P-gp. However, at the tested concentrations, we observed that the Aβ42-stimulated ATPase activity was less than half of the VPL-simulated ATPase activity. We note that this relationship may not extend across the entire possible concentration range. A possible explanation is that the large size of the Aβ42 peptide (4514 Da for Aβ42, versus 454.6 Da for VPL) may make it difficult for P-gp to move between structural states during the transport process. It is also possible that Aβ42 dissociates slowly from the DBDs due to strong hydrophobic interactions, thus explaining the lower stimulation of ATPase activity compared to VPL (Fig 4 and S2 Table). While increased stimulation of ATPase activity is considered a characteristic of P-gp substrates, it should be noted that some non-substrates can also stimulate the ATPase activity of P-gp [50, 51, 63].

To further test if P-gp can transport Aβ42, we performed fluorescence accumulation assays in a human cellular system. Our data show that inhibition of P-gp by the strong and P-gp specific inhibitor, tariquidar (TQR), resulted in increased intracellular accumulation of fluorescently labeled Aβ42 (HiLyte488-Aβ42). Interestingly, we observed this increased accumulation in both P-gp overexpressing DU145-TXR cells and the parental, non-P-gp overexpressing DU145 cells (Fig 4). Previous work by our group has shown that the chemotherapy-sensitive, non-P-gp-overexpressing DU145 cells do express detectable, but low, amounts of both P-gp and the Breast Cancer Resistance Protein (BCRP, *ABCG2*) [64]. While TQR is a strong inhibitor of P-gp, Kannan et al. have shown that TQR can act as an inhibitor of both P-gp and BCRP at concentrations greater than 100 nM [65]. However, the overexpression levels of P-gp and BCRP in DU145-TXR cells are significantly different–DU145-TXR cells overexpress P-gp much more than BCRP, relative to the parental DU145 cells [53, 64]. While evidence suggests that BCRP may play a role in the transport of both Aβ40 and Aβ42, the great difference between the levels of BCRP and P-gp overexpression in DU145-TXR cells suggests that inhibition of BCRP is unlikely to significantly affect our results [66, 67].

Interestingly, the uninhibited DU145-TXR cells exhibited significantly higher fluorescence than the uninhibited DU145 cells (Figs 4 and S5). We hypothesize that this is due to reduced CD33 levels in the DU145-TXR cell line (3.75 fold decrease relative to DU145 cells), a byproduct of generating resistance through exposure to the chemotherapeutic Paclitaxel [53]. CD33 is a transmembrane protein involved in cellular adhesion and Aβ clearance processes; reduced expression of CD33 has been shown to result in increased uptake of Aβ42 [68–70]. However, inhibition of the resistant DU145-TXR cells resulted in a 33% greater change in Aβ fluorescence relative to the change in DU145 cells, again strongly suggesting that inhibition of P-gp transport resulted in low levels of Aβ efflux through P-gp with a concomitant increase in Aβ accumulation in these cells.

## Conclusion

Through the combination of computational simulations, kinetic measurements of the purified protein, and transport assays in a human cellular environment, we have shown here that P-gp is able to transport the monomeric form of Aβ. While there is a growing body of evidence that P-gp plays an important role in the clearance of Aβ across the BBB, such conclusions are beyond the scope of this study [14, 16, 71]. Adding Aβ to the list of known substrates indicates that P-gp can transport much larger molecules than was previously thought. Given the clinical importance of P-gp and of other ABC transporters, we believe that the ability of human efflux pumps to transport large ligands, and the mechanism by which they do so, warrants further study.

## Materials and methods

### Docking Aβ to human P-glycoprotein (P-gp)

Aβ structures were docked to the drug binding domains (DBDs) of P-glycoprotein in the open-to-the-cytoplasm conformation (derived from the homologous 4KSB structure) of human P-gp using AutoDock Vina as described previously (S1 Fig) [31, 36, 37, 72]. Ligand interactions were limited to the cytoplasmic extensions of the transmembrane helices and the transmembrane sections of P-gp and used an exhaustiveness of 128 (the default exhaustiveness or number of replica docks for Vina is set at 8). Ligand binding to nucleotide binding domains (NBDs) was not investigated. The resultant ligand docking positions were ranked by predicted binding affinities; the conformational pose with the highest predicted affinity was used as a starting point for molecular dynamics (MD) simulations, except where indicated in the text.

Three different structures of Aβ were docked to the DBDs of P-gp. 2LFM, a partially folded solid state NMR structure of Aβ40 in an aqueous environment, docked with a predicted affinity of -7.2 kcal/mol (S2A Fig) [40]. 2M4J, a 40 residue Aβ fibril derived from AD brain tissue docked with a predicted affinity of -7.1 kcal/mol (S2B Fig) [38]. 1IYT, a solid state NMR structure of Aβ42 in an apolar microenvironment, with a predicted affinity of -7.2 kcal/mol (S2C Fig) [38]. As a control, every residue in the highest affinity docking pose of Aβ40 fibril 2LFM was mutated into an arginine, creating the Polyarginine 42 peptide (S2D Fig).

### Transport of Aβ through P-gp in molecular dynamics simulations

To facilitate the efflux of substrates across the cell membrane, ATP-driven ABC-transporter proteins undergo large conformational changes powered by the binding and hydrolysis of ATP. These conformational changes switch the drug binding domains (DBDs) of the transporter from "open to the cytoplasm" (inward facing) to "open to the extracellular space" (outward facing) [73, 74]. Such cycling has been hypothesized for P-gp and has previously been shown by us in computational simulations to transport small-molecule, drug-like ligands from the cytoplasmic membrane leaflet to the extracellular leaflet and extracellular space [35]. The modeled, putative catalytic cycle of P-gp therefore reflects the hypothesized sequence of conformational changes for ABC transporters and has allowed us to visualize substrate transport driven by P-glycoprotein. These previous studies have allowed us to investigate P-glycoprotein-driven movement of small drug-like molecules across the membrane. These computational simulations have been extended here to the larger, polypeptide substrates, Aβ40 and Aβ42.

To model a putative catalytic transport cycle of P-gp, we used crystal structures of P-gp homologues in various conformations as in McCormick et al. 2015 [35]. Because these structures were determined from crystals, the conformations represented by these structures are

relatively stable, representing relatively low energy conformations of the protein. During TMD simulations, small forces were applied to selected Cα atoms of P-gp to direct the movement of protein domains toward the respective target coordinates. The putative catalytic transport cycle of P-gp we used follows the sequence of conformational states based on earlier work [35, 36]: (1) a conformation with the DBDs wide open to the cytoplasm (derived from 4KSB); (2) a conformation with the DBDs slightly open to the cytoplasm (derived from 3B5X); (3) a conformation with fully engaged NBDs and DBD opened to the exterior (derived from 2HYD); and (4) a final conformation with NBDs in an ATP hydrolysis transition state and the DBDs fully open to the extracellular space (derived from 3B5Z) (S1 Fig). Using these TMD simulations, we guided P-gp through a putative transport cycle and included Aβ40 (2LFM, 2M4J) or Aβ42 (1IYT) in the DBDs on the cytoplasmic side of the membrane [36, 38–40].

The inward facing structure of the mouse P-gp (4KSB) has fully disengaged NBDs with its transmembrane DBDs oriented in an inward facing state [31]. The structure of MsbA from *Vibrio cholerae* (3B5X) has disengaged NBDs, and DBDs partially open to the cytoplasm. The structure of SAV1866 from *S. aureus* (2HYD) has engaged NBDs and its DBD open to the outside [73, 75]. The structure of MsbA from *S. typhimurium* (3B5Z) also has fully engaged NBDs with its DBD open to the outside but may represent a post-hydrolysis transition state since crystallization conditions included an MsbA—ADP-vanadate complex. Using the structures in the aforementioned sequence, we previously simulated the conformational changes of a putative catalytic transport cycle using models of human P-gp and targeted molecular dynamics (TMD) simulations [35, 36] based on structures from [76, 77].

### Preparation of the Aβ42 synthetic peptide

Previous cell culture studies of P-gp and Aβ42 produced conflicting results [12]. Additionally, Aβ42, specifically the oligomeric form, is associated with neurotoxic effects [78–80]. Thus, we sought to further explore, and hopefully elucidate, the relationship between P-gp and Aβ42 in our cell-based and biophysical assays.

### Monomerization of Aβ42 for ATPase assays.

The Aβ42 synthetic peptide was purchased from GenicBio Limited, PRC (sequence DAEFRHDSGYEVHHQKLVFFAEDVGSNKGAIIGLMVGGVVIA). The peptide had a molecular weight of 4514.14 g/mol and was judged to be 95.40% pure by HPLC. To monomerize the protein, lyophilized Aβ42 was removed from storage at -80˚C and allowed to equilibrate at room temperature for 30 minutes to avoid condensation upon opening the vial. In a fume hood, 1 mg of the lyophilized Aβ42 peptide was resuspended in 300 μl of 1,1,1,3,3,3-hexafluoro-2-propanol (HFIP, Sigma-Aldrich). The mixture was sonicated and vortexed thoroughly to ensure proper solvation. The solution was then centrifuged at 10,000 rpm for 5 min and the supernatant was moved to a clean tube. Centrifugation was repeated once more to remove any insoluble materials. The solution was then aliquoted into separate vials where the HFIP was allowed to evaporate in the fume hood overnight. The desiccated pellets were stored at -20˚C. For use in ATPase assays, the samples were resuspended in a 1:4 mixture of dimethyl sulfoxide (DMSO) and sterile water.

### Preparation of fluorescently labeled Aβ42 for cell culture assays.

Fluorescent (HiLyte™ Fluor 488) labeled Aβ42 was purchased for cell culture assays from Anaspec (HiLyte™ Fluor 488 –DAEFRHDSGYEVHHQKLVFFAEDVGSNKGAIIGLMVGGV-VIA). This peptide had a molecular weight of 4870.5 g/mol, absorption/emission wavelengths of 503/528 nm, and was judged to be > = 95% pure by HPLC (CAT# AS-60479, LOT#

1958003). To prepare the Aβ peptide for cell culture assays, 0.1mg of the peptide was dissolved in 30 uL of 1% (w/v) ammonium hydroxide in sterile water and filtered using a 0.45 um pore filter [81]. Once thoroughly dissolved and mixed, 380 uL of Phosphate Buffered Saline (PBS) solution were added to the peptide-ammonium hydroxide solution (final $NH_4OH$ 0.073%). After mixing thoroughly again, the peptide solution was aliquoted into 50 μL aliquots and frozen until use. When using the peptide solution for cell culture experiments, the peptide containing aliquot was thawed and vortexed immediately before use.

## Accumulation of fluorescently labeled Aβ42 in DU145 and DU145-TXR cells

Multidrug-resistant (MDR) DU145-TXR prostate cancer cells have been previously shown to overexpress P-glycoprotein [53]. These MDR DU145-TXR cells (kindly provided by Evan Keller, Univ. of Michigan) were derived from drug sensitive DU145 cancer cells by culturing in the presence of the chemotherapeutic paclitaxel to create the P-gp overexpressing cell line DU145-TXR [53]. Both DU145 and DU145-TXR cells were grown in complete media consisting of RPMI-1640 with L-glutamine, 10% fetal bovine serum, 100 U/mL penicillin and 100 μg/mL streptomycin in a humidified incubator at 37˚C using 5% $CO_2$. The drug-resistant line DU145-TXR was maintained under positive selection pressure by supplementing complete media with 10 nmol/L paclitaxel. Both cell lines were grown and seeded on collagen-treated flasks and plates (Collagen Type I, Corning).

To assess the accumulation of fluorescently labeled Aβ42 in both cell lines, cells were trypsinized from monolayers and seeded at 60,000 cells per well in 24 well plates in complete RPMI media. Prior to seeding, a sterilized glass coverslip was placed in each well, and each well was treated with a working solution of 0.01 mg/mL collagen Type I (Corning) in 0.02 N Acetic Acid for 10 minutes and rinsed with PBS. After 24 hours of incubation at 37˚C, the media was removed and replaced with fresh complete RPMI media. The cells were dosed with 1 μM of the P-gp inhibitor tariquidar (TQR), 1uM of HiLyte-488-Aβ42, a combination of both, or 2% DMSO media as a control, and incubated at 37˚C and 5% $CO_2$ for 16 hours [82].

We tested HiLyte-488-Aβ42 at a concentration of 1 μM for the following reasons: 1) at this concentration, HiLyte-488-Aβ42 was detectable without significant background fluorescence that would obfuscate quantification efforts; 2) the formation of HiLyte-488-Aβ42 fibrils was not observed at this concentration, and 3) our pilot studies observed toxicity effects against DU145-TXR cells at concentrations greater than 5 μM. The final concentration of ammonium hydroxide in each well was kept at approximately 0.0015% and was matched in controls to ensure identical treatment of all cell samples. If there was any leftover Aβ in the thawed aliquot, the excess Aβ was not re-frozen but was discarded to avoid aggregation that can be induced by freeze-thawing. After 16 hours of incubation, media was removed, and cells were gently washed with cold PBS. Cells were then fixed in 4% paraformaldehyde in PBS for 20 minutes. Cells were then stained with DAPI in PBS for 10 minutes, and subsequently washed twice with cold PBS. Each coverslip was removed from the well and mounted on a glass slide using Fluoromount G mounting fluid.

After drying slides overnight, fluorescence-confocal microscopy was performed on a Zeiss LSM800 microscope using Plan-Apochromat 20x and 40x/1.3 oil-immersion objectives. Fluorescence of HiLyte488-Aβ42 or DAPI was measured using the pre-programmed EGFP and DAPI filters and the Zeiss ZEN software, specifically ZEN 3.3. Specifically, the EGFP filter uses excitation/emission wavelengths of 488/509 nm, and DAPI uses excitation/emission of 353/465 nm. Detection wavelengths range from 495–565 nm for EGFP, and 410–470 nm for DAPI. We found that it was difficult to identify extracellular HiLyte488-Aβ42 at 20X magnification,

since extracellular HiLyte488-Aβ42 should not be included in the quantification of intracellular fluorescence. Therefore, we used the images taken at 40X to quantify HiLyte488-Aβ42 fluorescence, the inclusion of the Bright Field channel allowed us to identify extracellular HiLyte488-Aβ42 fluorescence.

Experiments were performed in duplicate, with two independent trials, and two samples per treatment in each trial. At 20X magnification, 12 images per individual slide were taken. At 40X magnification, 6 images per individual slide with HiLyte488-Aβ42 treatment were taken and used for quantification; at least three images were taken for TQR-only or DMSO control slides to confirm the lack of HiLyte488-Aβ42 fluorescence, which was confirmed initially by the images taken at 20X magnification. Thus for quantification of HiLyte488-Aβ42 fluorescence, data are presented as the mean and standard deviation, with 24 replicates per treatment, 12 replicates from each individual trial. All images were taken before analysis was performed; all images captured at 40X magnification of slides treated with HiLyte488-Aβ42 are included in the analysis. Z-stacks of the HiLyte488-Aβ42-treated cells, with or without TQR, were taken at 40X magnification to visualize any intracellular accumulation of HiLyte488-Aβ42. A set of Z-stacks were taken with the Bright Field channel to aid visualization of the DU145 or DU145-TXR cell boundaries. Z-stacks are shown in S1 Movie.

Quantification was performed using FIJI ((Fiji Is Just) ImageJ, NIH, Bethesda, Maryland, USA) [83–85] on raw, unedited CZI files. To quantify fluorescence with ImageJ, raw CZI image files were imported using the Bio-Formats plugin [86]. Analysis of HiLyte488-Aβ42 fluorescence with ImageJ was automated using ImageJ macros (included in Supplementary Files). Using a copy of the raw green channel image, regions of HiLyte488-Aβ42 fluorescence were "thresholded" as follows: (1) use the default "Subtract Background" function (rolling ball radius of 10px); (2) apply the default Unsharp Mask (radius 1 px, mask = 0.60 sigma) to define the edges of fluorescent areas; (3) threshold the image (minimum 15, maximum 255) to determine which areas will be measured; (4) convert the thresholded image to a binary mask; (5) save the binary mask as a .TIF image. Through this process, a binary mask is created in which areas of thresholded HiLyte488-Aβ42 fluorescence are black, and all other areas are white. The binary mask was then used to define regions for measurement on the original, raw green channel image. Using the binary mask to define regions for measurement, areas of thresholded HiLyte488-Aβ42 fluorescence in the raw green channel image were measured using the default "Analyze Particles" function of ImageJ (Supplementary Files). Using the previously created binary mask of each image, the mean background intensity of each raw green channel image was measured by selecting the inverse of the HiLyte488-Aβ42 areas, and then using the default "Measure" function. Aβ Fluorescence is quantified as a measure of the "corrected mean fluorescence intensity" using the following formulae and is reported in arbitrary units.

$$Integrated\ Density_{area} = Sum(value\ of\ each\ pixel\ in\ area) * 0.024329\ \mu m^2 pixel^{-1}$$

$$Mean\ Intensity = \frac{Integrated\ Density_{area}}{N_{pixels\ in\ area}}$$

$$Corrected\ Mean\ Intensity_{\alpha\beta} = Mean\ Intensity_{\alpha\beta} - Mean\ Intensity_{background}$$

Once the initial analysis was completed using ImageJ, images were examined for potential extracellular HiLyte488-Aβ42 fluorescence using the Bright Field channel overlaid with the green channel as a guide. In the event of extracellular particles, we (1) created a copy of the original binary mask for that image; (2) manually removed the extracellular particle from the binary mask copy; (3) ran the measuring and analysis functions using the 'corrected' binary

mask to redirect measurements. Any extracellular areas were manually removed from the binary mask, and the image was re-analyzed using the same macro. This procedure was performed for 10/48 total images; 6/11 were from HiLyte488-Aβ42 images; 4/11 were from HiLyte488-Aβ42 + TQR images. This resulted in an average change of 0.01 ± 0.42 a.u. from the original values. To compare the levels of HiLyte488-Aβ42 fluorescence between treatments, we report the Corrected Mean Intensity in arbitrary units. Data was analyzed using a two-tailed T test with equal variance and GraphPad Prism version 7 for Windows, GraphPad Software, San Diego, California USA, www.graphpad.com. The DAPI-GFP-BF merge images were enhanced for inclusion in figures, and viewing in a small format, using the Zen Blue 3.3 as follows–Bright field channel, levels adjusted using the 'Min Max' default setting to enhance contrast and cell outlines; DAPI, levels adjusted using the 'Best Fit' default setting to enhance contrast; GFP (Green channel for HiLyte488-Aβ42), levels adjusted by setting the maximum to 149. Zen Blue 3.3 was used to export high resolution files of each image for assembly into figures. Microscopy figures were assembled using Adobe Illustrator.

## ATPase activity assays with P-gp in nanodiscs and micelles

**Purification of murine P-gp.** Murine cys-less P-gp was used for all ATPase activity assays [45, 46, 87, 88]. Protein purification of P-gp and ATP hydrolysis assays were performed as described in Delannoy et al. 2005 and Brewer et al. 2014 with some modifications as described below [46, 87].

**Preparation of P-gp in micelles.** P-gp was expressed in *Pichia Pastoris* GS-115. To isolate P-gp from its native membrane and embed it in detergent micelles, 80 mL of frozen cell pellets were thawed in a 37˚C water bath, and protease inhibitors (160 μL pepstatin A, 32 μL Leucine, 16 μL chymostatin, 800 μL of 200 mM PMSF and 800 μL of 200 mM DTT) were subsequently added. Cells were then broken open using 175 mL of glass beads and a BeadBeater (Biospec products). The BeadBeater was filled to the top with buffer containing 30% glycerol, 50 mM Tris, 125 mM NaCl, 10 mM imidazole (pH 8.0). To prevent the samples from overheating during bead beating, ice with rock salt was used. Samples were spun at 10,000 rpm for 30 min at 4˚C to remove debris, nuclei, mitochondria and unbroken cells using a Beckman Avanti JXN-26 centrifuge. The supernatants were then subjected to a fast spin at 45,000 rpm for 45 min at 4˚C using Beckman Optima XPN-80. The resultant pellets (which contain P-gp) were washed with microsome wash buffer (20% glycerol, 50 mM Tris, pH 7.4), and resuspended in Tris buffer (30% glycerol, 50 mM Tris, 125 mM NaCl, 10 mM imidazole, pH 8.0). Nickel-NTA columns were used to capture P-gp engineered with His-tag.

Microsomes containing P-gp were diluted with Tris buffer (20% glycerol, 50 mM Tris, 50 mM NaCl, 10 mM imidazole, pH 8.0) to 2 mg/mL, and 0.6% n-Dodecyl β-D-maltoside (DDM, w/v) (Sigma-Aldrich) and 0.01% lysophosphatidylcholine (lyso-PC, w/v) were added. Then, the sample solution was sonicated in an ice-cold water bath (model) for 5 cycles of 5 min. Samples were then spun down at 20,000 rpm for 30 min at 4˚C using a Beckman Optima XPN-80 centrifuge to remove undissolved microsomal proteins. The supernatants were then applied to a Ni-NTA gravity column (QIAGEN), and incubated for 30 min at 4˚C. Flow-through was collected at 2 mL/min, and the column was washed with 20 bed volume of wash buffer (20% glycerol, 50 mM Tris, 50 mM NaCl, 20 mM imidazole, pH 7.5) at 1mL/min, followed by 10 bed volume of the second wash buffer (20% glycerol, 50 mM Tris, 50 mM NaCl, 40 mM imidazole, pH 7.5) at 1 mL/min. Protein was eluted with buffer (20% glycerol, 50 mM Tris, 50 mM NaCl, 300 mM imidazole, pH 7.5) for 3 bed volumes at 0.5 mL/min. Both buffers for the wash and elution were supplemented with 0.6% DDM (w/v) and 0.01% lyso-PC (w/v) if not otherwise

specified. The eluates were concentrated to about 150 μL using Amicon 100K centrifugation filters (MilliporeSigma) at 4˚C, and stored at -80˚C.

**Reconstitution of P-gp into nanodiscs.** P-gp was incubated with a 10x molar excess of membrane scaffold protein (MSP) and a 500x molar excess of 40% L-alpha-phosphatidylcholine (PC) from soybean (Sigma) for 1 hour at room temperature with gentle agitation to facilitate formation of nanodiscs [89]. Biobeads SM-2 (BioRad) were presoaked in methanol, washed with a large amount of water, equilibrated with equilibration buffer (20% glycerol, 50mM Tris-HCl pH 7.5, 50mM NaCl), and finally added at a ratio of 1.4g/mL to the assembly mix to remove detergent. After addition of Biobeads SM-2, the mixture was incubated for 1.5 hours at room temperature with shaking to remove detergent from the crude nanodisc sample. The Bio-beads were removed from the crude nanodiscs by piercing the bottom of the centrifuge tube with a 25 gauge needle and centrifuging at 1000xg for 1 minute. Empty discs were removed by Ni-NTA column chromatography, utilizing the histidine–tag at P-gp. Six bed volumes of col-umn wash buffer (20% glycerol (v/v) 50mM Tris-HCl pH 7.5 at RT, 50mM NaCl, 20mM imid-azole) were then applied. Purified nanodiscs were eluted from the column by applying 1 bed volume elution buffer (20% (v/v) glycerol, 50mM Tris-HCl pH 7.5, 4˚C, 50mM NaCl, 300mM imidazole). Samples were analyzed by gradient SDS-PAGE and coupled enzyme assays.

## ATPase activity assays

Briefly, ATP hydrolysis by P-gp was coupled to the oxidation of NADH to NAD$^+$ by two enzymes, pyruvate kinase and lactate dehydrogenase, as described in [46]. The coupled enzyme assay cocktail included 50 mM Tris, pH 7.5, 24 mM MgSO$_4$, 20mM KCl, 1.94 mM phospho-enolpyruvate) (PEP), 0.058 mg/mL pyruvate kinase, 0.0288 mg/mL lactate dehydrogenase, 1.13 mM NADH, and 4 mM ATP. The absorbance decrease of NADH at 340 nm was recorded using a BioTek Eon plate reader BioTek. The ATPase activity of P-gp was directly correlated with the rate of NADH oxidation. An extinction coefficient of 6220 M$^{-1}$cm$^{-1}$ at 340 nm was used for calculations of NADH oxidation with a measured path length of 0.6 cm. Aβ42 was incubated with P-gp for 30 min at 37˚C before the addition of coupled enzyme assay cocktail.

In the activity assays reported here, 15 μg of purified P-gp was used for nanodiscs, and 20 ug of purified P-gp was used for micelles. A molar ratio of 1:18, P-gp:Aβ42, was used for exper-iments with both nanodiscs and micelles—for P-gp in mixed micelles, 712 nM P-gp to 12.8 μM Aβ42 was used, and for P-gp in nanodiscs, 534 nM of P-gp to 9.6 μM of Aβ42 was used. The basal ATPase activity of P-gp in mixed micelles was 51 ± 3 nmol/min/mg, and the Verapamil (VPL)-stimulated activity was 106 ± 7 nmol/min/mg. The basal ATPase activity of P-gp in nanodiscs was 131 ± 9 nmol/min/mg, and the VPL-stimulated activity was 390 ± 14 nmol/min/mg. A VPL concentration of 150 μM was used, as in [49]. ATPase activity experi-ments to measure basal and stimulated activity were performed with n = 4. For ATPase activity experiments with VPL and/or Aβ42, experiments were performed with n = 3. One protein preparation was used for each of the different conditions (micelles and nanodiscs). Resultant data represent the mean ± one standard error of the mean.

## Supporting information

**S1 Fig. Conformational states of P-glycoprotein used in targeted molecular dynamics sim-ulations.** Targeted Molecular Dynamics (TMD) simulations were performed as described in McCormick et al. 2015 using a dynamic model of human P-glycoprotein and several low energy conformations of P-gp homologues [35, 36]. Here we show the target names of the homologs and corresponding conformation of P-gp for each step in the putative catalytic cycle. (TIF)

**S2 Fig. Aβ or control peptides at the first and final frames of targeted molecular dynamics experiments.** (A) Aβ40 (2LFM), (B) Aβ40 (2M4J), (C) Aβ42 (1IYT), (D) Polyarginine 42. Panels A-D are representative images of bound peptides at the start of molecular dynamics simulations. (A) 1IYT docked to the DBDs of P-gp with an estimated affinity of -7.2 kcal/mol. 1IYT is a solid state NMR structure of Aβ42 solved in an apolar microenvironment [38]. (B) 2LFM is a partially folded Amyloid-β (Aβ) 40 structure that was solved in an aqueous environment [40]. 2LFM docked to the drug binding domains (DBDs) of P-glycoprotein (P-gp) with a predicted affinity of -7.2 kcal/mol. (C) 2M4J is an Aβ40 fibril derived from brain tissue with Alzheimer's disease [39]. 2M4J docked to the DBDs of P-gp with an estimated affinity of -7.1 kcal/mol. Panels E-H are representative images at the end of a TMD simulation of the putative catalytic cycle. The bound peptide is shown in purple licorice representation or in surface representation colored for lipophilicity (teal = hydrophilic, gold = lipophilic, and white = neutral). The N- and C-terminal halves of P-gp are colored turquoise or orange.
(TIF)

**S3 Fig. Total movement of amyloid-β, polyarginine 42, or daunorubicin through P-gp during a putative catalytic cycle.** The center of mass of each ligand was calculated for each step of the simulations relative to the distance from the starting location in the X, Y, and Z axes. The plane of the membrane is parallel to the X and Y plane; movement through the membrane is oriented on the Z-axis. Distances are presented in Å. Six simulations were performed for each ligand; data represent the mean total distance ± one standard deviation from the mean. *Data for daunorubicin is reproduced with permission from *McCormick et al. 2015* [35]. Copyright 2015 American Chemical Society.
(TIF)

**S4 Fig. The accumulation of HiLyte488-Aβ42 in DU145 and DU145-TXR cancer cells.** The intracellular fluorescence of paired chemotherapeutic sensitive/resistant cancer cell line (DU145 and DU145-TXR) was measured by confocal microscopy after incubation with 1 μM fluorescently labeled Aβ42 in the presence or absence of 1 μM Tariquidar (TQR). Statistical significance was determined using an unpaired T-test in Graphpad Prism; data are n = 24 images per treatment, two trials per treatment. Data are expressed as arbitrary units (a.u.) as calculated by the Integrated Density function of ImageJ [83–86].
(TIF)

**S5 Fig. Distance between LYS28 and ASP23 of Aβ monomers during simulated catalytic transport cycles of P-gp.** The distance in Angstroms between the charged nitrogen (N) of LYS28 and either potentially charged oxygen (OD1, or OD2) of ASP23 in the respective Aβ peptides. Panels (A-C) show the position of K28 and D23 in each Aβ monomer. Graphs (D,E) show the mean distance in Angstroms between K28 and D23 in Aβ 42 (1IYT) during TMD simulations; graphs (F,G) show the mean distance in Angstroms between K28 and D23 in Aβ 40 (2LFM); graphs (H,I) show the mean distance in Angstroms between K28 and D23 in Aβ 40 (2M4J). Data represent the mean distance between the two selected atoms ± one standard deviation shown in colored shading (n = 6).
(TIF)

**S6 Fig. The accumulation of HiLyte488-Aβ42 in DU145 cancer cells.** The intracellular fluorescence of chemotherapeutic sensitive DU145 cells was measured by confocal microscopy. Fluorescence was measured 24 hours after a 16 hour incubation with fluorescently labeled Aβ42 monomers. Representative Images (A-D) show DU145 treated with DMSO alone; representative images (E-H) show DU145 treated with 1 μM Tariquidar (TQR) alone; representative

images (I-L) show DU145 treated with 1 μM Aβ42 alone; representative images (M-P) show DU145 treated with 1 μM Aβ42 and 1 μM TQR. Data are n = 12 images per trial, two trials.
(TIF)

**S7 Fig. The accumulation of HiLyte488-Aβ42 in DU145-TXR P-gp overexpressing cancer cells.** The intracellular fluorescence of chemotherapeutic resistant, P-gp overexpressing DU145-TXR cells was measured by confocal microscopy. Fluorescence was measured 24 hours after a 16 hour incubation with fluorescently labeled Aβ42 monomers. Representative Images A-D show DU145-TXR treated with DMSO alone; representative images E-H show DU145-TXR treated with 1 μM Tariquidar (TQR) alone; representative images I-L show DU145-TXR treated with 1 μM Aβ42 alone; representative images M-P show DU145-TXR treated with 1 μM Aβ42 and 1 μM TQR. Data are n = 12 images per trial, two trials.
(TIF)

**S8 Fig. Number and characteristics of residue contacts made by Aβ42 peptides to the drug binding domains of P-gp during TMD simulations.** Data are reported as the number and type of residues in the P-gp Drug Binding Domain with an α-carbon within 3 Å of the respective Aβ peptide. Residues are classified as Polar (SER, THR, CYS, ASN, GLN, TYR), Non-Polar (GLY, ALA, VAL, LEU, MET, ILE, PHE, PRO, TRP), Positively Charged (LYS, ARG, HIS), or Negatively Charged (GLU, ASP). Residues were included in the total count if the respective Aβ peptide interacted with the residue in 4/6 TMD simulations.
(TIF)

**S1 Movie. Intracellular accumulation of Hylyte-488-AB42 with or without the P-gp inhibitor tariquidar.** Z-stacks videos shown at 40x-magnification.
(MP4)

**S1 Table. Number and type of residue contacts made by Aβ monomers with the drug binding domains of P-gp during TMD simulations.**
(DOCX)

**S2 Table. Movement of Aβ and daunorubicin by P-gp according to structural transition.**
(DOCX)

**S3 Table. Mean brightness of Aβ42 fluorescence intensity in cell culture experiments.**
(DOCX)

## Acknowledgments

The authors would like to acknowledge and thank Kelsey Paulhus for her valuable and constructive suggestions and guidance during the planning and development of the microscopy in this work. We would also like to thank Amila K. Nanayakkara for his valuable contributions he made on a pilot study for the accumulation assays in this work. We would like to thank Dr. Kimberly Reynolds (UT Southwestern Medical Center, Dallas) for reviewing the manuscript and for her advice. We would also like to thank Professor Heng Du (University of Texas, Dallas) for advice and methods for handling Amyloid β preparations and Professor Evan Keller (U. of Michigan) for the DU145 and DU145-TXR cell lines. We would like to thank Ms. Suzy Ruff for her support of our work.

## Author Contributions

**Conceptualization:** John G. Wise.

**Data curation:** James W. McCormick, Lauren A. McCormick, Gang Chen, John G. Wise.

**Formal analysis:** James W. McCormick, Lauren A. McCormick, Pia D. Vogel, John G. Wise.

**Funding acquisition:** John G. Wise.

**Investigation:** James W. McCormick, Lauren A. McCormick, Gang Chen, Pia D. Vogel, John G. Wise.

**Methodology:** James W. McCormick, Lauren A. McCormick, Gang Chen, John G. Wise.

**Resources:** Lauren A. McCormick, John G. Wise.

**Software:** James W. McCormick, John G. Wise.

**Supervision:** John G. Wise.

**Validation:** James W. McCormick, Lauren A. McCormick.

**Visualization:** James W. McCormick.

**Writing – original draft:** James W. McCormick.

**Writing – review & editing:** James W. McCormick, Lauren A. McCormick, Pia D. Vogel, John G. Wise.

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
