## [Decision Letter · Decision Letter 0]

25 Jan 2021

PONE-D-20-39570

Transport of Alzheimer’s associated amyloid-β catalyzed by P-glycoprotein

PLOS ONE

Dear Dr. Wise,

Thank you for submitting your manuscript to PLOS ONE. After careful consideration, we feel that it has merit but does not fully meet PLOS ONE’s publication criteria as it currently stands. Therefore, we invite you to submit a revised version of the manuscript that addresses the points raised during the review process.

In particular, both reviewers raised a similar concern that the proper literatures to be cited or referred are lacking in the current form, so the suitable and respectable responses should be required.

We look forward to receiving your revised manuscript.

Kind regards,

Kazuma Murakami, Ph.D.

Academic Editor

PLOS ONE

Journal Requirements:

Reviewers' comments:

Reviewer's Responses to Questions

**Comments to the Author**

1. Is the manuscript technically sound, and do the data support the conclusions?

Reviewer #1: Partly

Reviewer #2: Partly

2. Has the statistical analysis been performed appropriately and rigorously? 

Reviewer #1: Yes

Reviewer #2: Yes

3. Have the authors made all data underlying the findings in their manuscript fully available?

Reviewer #1: Yes

Reviewer #2: Yes

4. Is the manuscript presented in an intelligible fashion and written in standard English?

Reviewer #1: Yes

Reviewer #2: Yes

5. Review Comments to the Author

Reviewer #1: The authors have tackled the important issue of P-gp mediated transport of amyloid peptides and the relationship to development of neurodegenerative diseases. They have highlighted earlier studies that fuelled some controversy. However, the published article arguing against this proposal stemmed from an inability to reproduce observations by other groups. There are now a decent number of publications promoting the P-gp involvement. Two further publications were released in 2020 and actually support the current manuscript – these should be acknowledged. Together, this will bring an important issue to a partial conclusion and then allow the field to move towards overcoming the issue of aberrant P-gp activity. Excellent structural work highlighting the binding sites on P-gp should also be cited – and there are many other instances of reference neglect in this submission. The authors need to credit related work and use it to support their case.

I am supportive of this article, but have also identified a number of issues that need to be resolved, as detailed in the points below.

Major Points:

1. An article in 2020 (Mamma Mia, P-glycoprotein binds again. FEBS Lett. 2020 Oct 6. doi: 10.1002/1873-3468.13951. Epub ahead of print. PMID: 33022784.) provided molecular docking data on the interaction of both Aβ40 and Aβ42 peptide with Pgp. A number of points related to this interaction were raised in the earlier work and this should be cited (also an issue with the ATPase data – see below). How do those docking data compare with the MD simulations in this manuscript? These articles come up with similar conclusions, so it is in the author’s best interests to use them as supporting evidence! In addition, a further structural study (Cryo-EM structures reveal distinct mechanisms of inhibition of the human multidrug transporter ABCB1. Proc Natl Acad Sci U S A 117, 26245-26253 (2020)) provides significant information on residues involved with substrate binding. Does the docking data in the current manuscript reconcile with this reference? A comment is required on this important binding issue.

2. One of the key points raised in the earlier manuscript was the potential difficulty in a conventional alternating access model being able to move Aβ peptides. It was suggested that the peptides might be “pinched” during the transport cycle. The authors should comment on how they see the large peptide traversing the cavity in what is regarded as quite a tight fit.

3. The absolute rate of basal and maximal ATPase activity must be provided – i.e. in units of µmol nucleotide/min/mg protein (or µmol phosphate/min/mg protein). This will enable comparison with the generally accepted values for purified Pgp and provide confidence in the protein preparations. It is fine to have relative or normalised values, but at some juncture, the absolute rates are essential.

4. What was the concentration of Aβ peptide used in the ATPase assays. Only a molar ratio is provided. The value in molar units is required and I am surprised that it was not included, specifically in the ATPase data.

5. A consistent drawback in the manuscript is that only a single concentration of substrates, ligands etc. were used in the functional assays. This prevents assessment of affinity and provision of the true extent of effects. The authors must indicate why specific concentrations of verapamil and Aβ peptide were used. The verapamil concentration of 150µM is exceedingly high and the conventional value used is in the range 10-30µM. Often higher concentrations of this compound can actually reduce the rate of ATP hydrolysis.

6. A recent manuscript (New Evidence for P-gp-Mediated Export of Amyloid-β Peptides in Molecular, Blood-Brain Barrier and Neuronal Models. International Journal of Molecular Sciences. 2021; 22(1):246.) demonstrated a dose-dependent effect of two Aβ peptides on ATPase activity of Pgp in liposomes. The extent of their stimulation was considerably higher than in the current paper. Potentially it is a concentration issue. The earlier study should be cited, hydrolysis rates (in absolute values) compared and a reason for the discrepancy offered.

7. On page 9 (line 168) the authors claim that stimulation of ATPase activity renders a compound as a transported substrate. This does not hold true for many compounds - many inhibitors stimulate hydrolysis. Yes, it is suggestive, but the text should be toned down.

8. It was very difficult to find the molecular identity of the fluorescent Aβ-peptide. The fluorescent moiety is “Hylyte488-Aβ42” and this should be used throughout the manuscript and the "fl-Aβ" replaced. A fluorescent tag on the peptide is not the same as the native peptide after all and this distinction should be offered.

9. I cannot find the wavelengths used for the fluorescent analyses. Figure 4 contains a clue – GFP. Was the wavelength/filter set for this fluorophore. Several of the methods provided exquisite detail and then missed key points like this.

10. The authors need to comment on why there is a Tariquidar effect on in the parental cell line (DU-145) in Figure 4 given that there is no Pgp? Tariquidar is known to affect ABCG2/BCRP – is this an explanation?

11. On first glance, the error bars in Figure 4 do not suggest a significant difference. But the authors have used an apparent n=24, which facilitated reaching significance. That is a little disingenuous, since the data were actually taken from two separate sample preparations. The n=24 is essentially an internal replicate. The authors might consider sampling a minimum of three separate (or independent) sections/samples in order to provide a true value of the assay variability.

12. Figure 4 contains a single concentration of Aβ peptide (1µM). Why was this value chosen – it needs to be justified.

13. On page 13, line 246 the authors suggest that Aβ peptide stimulates the ATPase activity of P-gp to a lesser extent than verapamil. That cannot be claimed when using a single concentration of each compound! At those two concentrations there is a difference, but one cannot say that the extent of stimulation will follow this order throughout a concentration range.

14. The point that the ATPase activity is lower in micelles than membranes (i.e. nanodiscs here) is certainly not a new one. It may be argued that due to the rigid environment of a nanodisc, that the peptides will preferentially enter the micelle environment to stimulate P-gp activity. Intercalation in a micelle will be energetically more favourable than a bilayer for the peptide.

Minor Points:

1. The introduction section could, and should, be trimmed by one third. It contains too much fundamental information simply not required for the specialist readership.

2. Page 3, line 15 – the build-up of amyloid deposits is actually an imbalance in the relative extents of formation and clearance. It cannot be exclusively attributed to clearance.

3. Simply express p-values as <0.05, 0.01 etc. There is no statistical justification for extended values such as p<0.0079. Recall that this is a dimensionless unit and merely a probability factor.

4. In Figure 4a, the label comprises DU145 (Pgp wild type). This is ambiguous and very confusing for the uninitiated. Wild type may mean P-gp with no mutations! I assume the authors mean that it is the wild type cell line with NO P-gp(?) Perhaps label it as DU145 (sensitive) or (resistant).

5. The MD assay to measure transport provided some interesting data and discussion points. However, the apparent transport of Aβ peptide should be referred to as “simulated movement” given that it is a theoretical data set.

6. The correct term (page 13 line 245) is dissociates and not “disassociates” – please replace here and in any other lines of the text.

Reviewer #2: McCormick and authors use a few different techniques (MD simulations, P-gp purified protein and cell studies overexpressing the transporter) to provide further evidence that P-gp is involved in the export of amyloid peptides. The data are of interest but the authors have not quoted recent literature in this area or older studies for that matter and need to give proper acknowledgement to the large body of evidence that is out there. For example, the study by Lam (2001) is quoted but not given proper acknowledgement and discussion of the data generated in this groundbreaking study that put P-gp on the map as a transporter of amyloid peptides. Also, a significant number of studies address the involvement of P-gp at the BBB endothelium and warrant proper acknowledgement and discussion (not just quoting).

Further specific points:

1. MD simultations: The authors have picked a polyarginine “control” peptide that is indeed similar in size and weight to 1-42 but very different in charge, therefore not being the best control here and not surprising that it wasn’t transported. Please comment.

2. A paper by Callaghan and colleagues (FEBS Lett Oct 2020) provides docking models of P-gp and a-beta peptides. Please comment on these authors observations.

3. Why was murine P-gp chosen for the ATPase activity studies, considering that rodents express multiple isoforms ? And why was only 1-42 included in these experiments and 1-40 left out ?

4. On Page 4, the authors compare the ATPase activity of P-gp reconstituted in micelles vs nanodiscs and conclude that the difference in findings is due to difference in membrane environment. Considering that they are comparing two different ‘in vitro’ methods, all with their pro’s and con’s as well as different incorporation efficiencies, this conclusion needs to be toned down as it is highly speculative and not based on much. Also, a recent paper by Chai et al (IJMS Dec 2020) shows that both 1-40 and 1-42 peptides stimulate ATP hydrolysis of human P-gp. Please discuss the results in light of your own findings.

5. Experiments looking at the accumulation of labelled 1-42 peptides in cell lines, again why was 1-40 not included ?

6. The images in Figure 4 are substandard and the authors need to provide better images.

7. If the TXR cells in Figure 4 are overexpressing P-gp, wouldn’t one expect that they have much less 1-42 (i.e. GFP) signal as they keep the amyloid peptide out ? Why then is the GFP image for the DU145 vs DU145-TXR (untreated with the P-gp inhibitor) in both cases similar in signal quantity ?

6. PLOS authors have the option to publish the peer review history of their article (what does this mean?). If published, this will include your full peer review and any attached files.

Reviewer #1: No

Reviewer #2: No

---

## [Author Response · Author response to Decision Letter 0]

2 Mar 2021

Please see the "Response to Reviewers.docx" uploaded in the revision process.

---

## [Decision Letter · Decision Letter 1]

23 Mar 2021

PONE-D-20-39570R1

Transport of Alzheimer’s associated amyloid-β catalyzed by P-glycoprotein

PLOS ONE

Dear Dr. Wise,

Thank you for submitting your manuscript to PLOS ONE. After careful consideration, we feel that it has merit but does not fully meet PLOS ONE’s publication criteria as it currently stands. Therefore, we invite you to submit a revised version of the manuscript that addresses the points raised during the review process.

We look forward to receiving your revised manuscript.

Kind regards,

Kazuma Murakami, Ph.D.

Academic Editor

PLOS ONE

Journal Requirements:

Reviewers' comments:

Reviewer's Responses to Questions

**Comments to the Author**

1. If the authors have adequately addressed your comments raised in a previous round of review and you feel that this manuscript is now acceptable for publication, you may indicate that here to bypass the “Comments to the Author” section, enter your conflict of interest statement in the “Confidential to Editor” section, and submit your "Accept" recommendation.

Reviewer #1: (No Response)

Reviewer #2: All comments have been addressed

2. Is the manuscript technically sound, and do the data support the conclusions?

Reviewer #1: Yes

Reviewer #2: Yes

3. Has the statistical analysis been performed appropriately and rigorously? 

Reviewer #1: Yes

Reviewer #2: Yes

4. Have the authors made all data underlying the findings in their manuscript fully available?

Reviewer #1: Yes

Reviewer #2: Yes

5. Is the manuscript presented in an intelligible fashion and written in standard English?

Reviewer #1: Yes

Reviewer #2: Yes

6. Review Comments to the Author

Reviewer #1: The revised article has addressed the vast majority of the points raised in the first round. It has been substantially improved; however, there are some minor adjustments that are needed as indicated below.

Point 1 (ATPase activity)

The values for ATP hydrolysis are a little on the low side, but within the acceptable range. The authors provided the absolute values in the revised version but have done so in the methods section. These are quite clearly results and should be placed in the section of results dealing with ATP hydrolysis.

Point 2 (peptide concentrations)

The molar concentrations should be added to the legend for Figure 3 and so should the values for verapamil.

Point 3 (high verapamil concentrations)

The first sentence of the rebuttal (and the citation to Lerner-Marmarosh) should be added to the results section on ATP hydrolysis.

Point 7 (using only peptide 1-42)

The justification is acceptable and I would recommend adding the statement on the use of 1-42 peptide since it is the more neurotoxic should be added to the manuscript – potentially in the methods or results section.

Point 11 (size of peptide)

The author’s rebuttal is well argued and raises many important points. The issue of substrate size and the perceived inability of Pgp to transport the peptides is a common argument in the field. It is used frequently to dismiss the involvement of Pgp in amyloid peptide removal – despite the considerable evidence. Consequently, I would advise an abbreviated version of this rebuttal be inserted into the manuscript. This would provide significant added value to the community.

Point 17 (fluorescent peptide concentration)

The justification for using 1uM fluorescent peptide is well made – please insert a statement in the methods section of the manuscript.

Reviewer #2: The authors have addressed all requests by reviewers and we congratulate them on an excellent publication

7. PLOS authors have the option to publish the peer review history of their article (what does this mean?). If published, this will include your full peer review and any attached files.

Reviewer #1: No

Reviewer #2: No

---

## [Author Response · Author response to Decision Letter 1]

31 Mar 2021

(Same content as "Response to Reviewers2.docx)

Reviewer #1: The revised article has addressed the vast majority of the points raised in the first round. It has been substantially improved; however, there are some minor adjustments that are needed as indicated below.

Reviewer #2 – all issues were addressed

Point 1 (ATPase activity): The values for ATP hydrolysis are a little on the low side, but within the acceptable range. The authors provided the absolute values in the revised version but have done so in the methods section. These are quite clearly results and should be placed in the section of results dealing with ATP hydrolysis.

We have updated the results section on page 8 to include the following statement:

“In micelles, we report that the basal activity of P-gp was 51 ± 3 nmol/min/mg, and the VPL-stimulated activity was 106 ± 7 nmol/min/mg. In nanodiscs, we report that the basal activity of P-gp was 131 ± 9 nmol/min/mg, and the VPL-stimulated activity was 390 ± 14 nmol/min/mg.”

Point 2 (peptide concentrations): The molar concentrations should be added to the legend for Figure 3 and so should the values for verapamil.

We have updated the legend of Figure 3 to include this important information. The following sentences were added in response to this request:

“For P-gp in micelles, 12.8 µM Aβ42 and 712 nM P-gp were used; for P-gp in nanodiscs, 9.6 µM Aβ42 and 534 nM P-gp were used – both corresponding to a molar ratio of 1:18. A VPL concentration of 150 µM was used, as in [51].”

Point 3 (high verapamil concentrations)- The first sentence of the rebuttal (and the citation to Lerner-Marmarosh) should be added to the results section on ATP hydrolysis.

We agree with this suggestion, and we have added the sentence – and accompanying citation - to the ATPase activity of P-gp in the presence of AB42 section of Results (page 8):

“A concentration of 150 µM verapamil was selected as this was the amount previously used to obtain maximal activity for murine P-gp expressed in Pichia pastoris as reported in Lerner-Marmarosh et al [49].” 

Point 7 (using only peptide 1-42)- The justification is acceptable and I would recommend adding the statement on the use of 1-42 peptide since it is the more neurotoxic should be added to the manuscript – potentially in the methods or results section.

We agree with this suggestion. We have added the following statement under the “preparation of the AB42 synthetic peptide” section in Methods; references correspond to the bibliography:

“Previous cell culture studies of P-gp and Aβ42 produced conflicting results [12]. Additionally, Aβ42, specifically the oligomeric form, is associated with neurotoxic effects [76-78]. Thus, we sought to further explore, and hopefully elucidate, the relationship between P-gp and Aβ42 in our cell-based and biophysical assays. 

Point 11 (size of peptide)- The author’s rebuttal is well argued and raises many important points. The issue of substrate size and the perceived inability of P-gp to transport the peptides is a common argument in the field. It is used frequently to dismiss the involvement of P-gp in amyloid peptide removal – despite the considerable evidence. Consequently, I would advise an abbreviated version of this rebuttal be inserted into the manuscript. This would provide significant added value to the community.

Original comment (for our reference): “One of the key points raised in the earlier manuscript was the potential difficulty in a conventional alternating access model being able to move Aβ peptides. It was suggested that the peptides might be “pinched” during the transport cycle. The authors should comment on how they see the large peptide traversing the cavity in what is regarded as quite a tight fit.

We agree with this suggestion. We have inserted an abbreviated version of our rebuttal into the appropriate sections of the discussion. Below, we have copied the original discussion text, and then the revised discussion text below. Text in italics was inserted or changed in response to this request from our reviewers. We feel that these additions significantly enhance the discussion and fully cover the requested changes. We note that we also discuss the docking results in the Results section on p 5-6, lines 108-117. Full changes are tracked on our re-submitted version of the paper.

Original Discussion (p11-12) ----------------------------------------

The mechanism by which P-gp might transport substrates of such significant size and flexibility as the Aβ peptides remains unclear. Despite being particularly suited to exploring problems of this nature, MD simulations of AB and P-gp have not been performed. Using previously established techniques, we explored how P-gp might transport Aβ40 and Aβ42 using targeted MD simulations.

In each simulation of the Aβ peptides, we observed vectorial movement of Aβ through the P-gp DBDs and towards the extracellular space, with total movement ranging between 7.8 and 9.4 Å (Figs 2 and S3). These distances correlate well with previously published movements of the P-gp substrate daunorubicin (DAU) in TMD simulations [35]. Interestingly, for both Aβ42 and the 2M4J structure of Aβ40, the bulk of observed movement occurred during the transition between the 3B5X and 2HYD conformations of P-gp, or when the DBDs switch from open-to-the-inside to open-to-the-outside (S1 Fig, S1 Table). In contrast, for both DAU and the 2LFM structure of Aβ40, the bulk of observed movement occurred during the transition from 2HYD to 3B5Z, both of which are open-to-the-outside conformations (S1 Table). 

Studies have shown that Aβ monomers can fold into structures with two β strands; these β strands allow the monomers…[Discussion section continues].

Edited Discussion (p11-12) --------------------------------------

The mechanism by which P-gp might transport substrates of such significant size and flexibility as the Aβ peptides remains unclear. Specifically, two primary questions remain: firstly, can the Aβ peptides fit within the DBDs of P-gp at all, and secondly, are the alternating access motions of P-gp sufficient to move the Aβ peptides across the cell membrane [8]? In answer to the first open question, docking studies performed by us and by Callaghan et al. show that the DBDs of P-gp can accommodate the Aβ peptides in both ordered and disordered states, thus the initial association of Aβ with the DBDs should be possible [42]. To address the second question, we explored how P-gp might transport Aβ40 and Aβ42 using targeted MD simulations and previously established techniques. Our TMD simulations showed that P-gp is indeed capable of transporting the Aβ peptides through the membrane despite their significant sizes.

In each simulation of the Aβ peptides, we observed vectorial movement of Aβ through the P-gp DBDs and towards the extracellular space, with total movement ranging between 7.8 and 9.4 Å (Figs 2 and S3). These distances correlate well with previously published movements of the P-gp substrate daunorubicin (DAU) in TMD simulations [35]. Interestingly, for both Aβ42 and the 2M4J structure of Aβ40, the bulk of observed movement occurred during the transition between the 3B5X and 2HYD conformations of P-gp, or when the DBDs switch from open-to-the-inside to open-to-the-outside (S1 Fig, S1 Table). In contrast, for both DAU and the 2LFM structure of Aβ40, the bulk of observed movement occurred during the transition from 2HYD to 3B5Z, both of which are open-to-the-outside conformations (S1 Table). 

As postulated by Callaghan et al., we observed that the Aβ peptides were indeed ‘pinched’ during simulated transport by P-gp [42]. However, in our TMD simulations, we found that the transported Aβ peptides were in a more disordered state than that postulated by Callaghan et al. This is unsurprising, since Aβ is partially exposed to the solvent when bound to the inward-open conformation of P-gp, and the helical structures of Aβ have been shown to collapse in water [54]. Since the Aβ peptides are thought to associate with P-gp though a complex network of interactions [8, 20, 55], it is possible that the peptides would contact the solvent prior to entering the DBDs of P-gp, and therefore arrive at the pump in a more disordered state. The partial – or complete – collapse of the ordered helices could further enhance the fit and subsequent transport of these bulky Aβ peptides by P-gp.

Studies have shown that Aβ monomers can fold into structures with two β strands; these β strands allow the monomers…[Discussion section continues]

Point 17 (fluorescent peptide concentration): The justification for using 1uM fluorescent peptide is well made – please insert a statement in the methods section of the manuscript.

We thank the reviewers for their assessment of our methods. We have inserted the following statement in the Accumulation of fluorescently labeled Aβ42 in DU145 and DU145-TXR Cells section of Methods (Page 19):

“We tested HiLyte-488-Aβ42 at a concentration of 1 µM for the following reasons: 1) at this concentration, HiLyte-488-Aβ42 was detectable without significant background fluorescence that would obfuscate quantification efforts; 2) the formation of HiLyte-488-Aβ42 fibrils was not observed at this concentration, and 3) our pilot studies observed toxicity effects against DU145-TXR cells at concentrations greater than 5 µM.”

---

## [Editor Report · Decision Letter 2]

6 Apr 2021

Transport of Alzheimer’s associated amyloid-β catalyzed by P-glycoprotein

PONE-D-20-39570R2

Dear Dr. Wise,

We’re pleased to inform you that your manuscript has been judged scientifically suitable for publication and will be formally accepted for publication once it meets all outstanding technical requirements.

Kind regards,

Kazuma Murakami, Ph.D.

Academic Editor

PLOS ONE
---

## [Editor Report · Acceptance letter]

14 Apr 2021

PONE-D-20-39570R2 

Transport of Alzheimer’s associated amyloid-β catalyzed by P-glycoprotein 

Dear Dr. Wise:

I'm pleased to inform you that your manuscript has been deemed suitable for publication in PLOS ONE. Congratulations! Your manuscript is now with our production department. 

Kind regards, 

on behalf of

Dr. Kazuma Murakami 

Academic Editor

PLOS ONE